# Comparative Analysis of Embryonic Development and Mitochondrial Genome of a New Intergeneric Hybrid Grouper (*Epinephelus fasciatus* ♀ × *Plectropomus leopardus* ♂)

**DOI:** 10.3390/ani15233445

**Published:** 2025-11-28

**Authors:** Xinlu Jiao, Tangtang Ding, Yongsheng Tian, Yongjun Guo, Yimeng Wang, Shihao Wang, Chunbai Zhang, Fengfan Yang, Linna Wang, Zhentong Li, Linlin Li, Yidan Xu, Yang Liu

**Affiliations:** 1Aquaculture Department, Fisheries College, Tianjin Agricultural University, Tianjin 300392, China; 15612990136@163.com (X.J.); dingtangtang0928@163.com (T.D.); guoyongjun@tjau.edu.cn (Y.G.); 18222630398@163.com (Y.W.); 2State Key Laboratory of Mariculture Biobreeding and Sustainable Goods, Yellow Sea Fisheries Research Institute, Chinese Academy of Fishery Sciences, Qingdao 266071, China; tianys@ysfri.ac.cn (Y.T.); wangsh9899@163.com (S.W.); zhangchunbai204@163.com (C.Z.); yangfengfan2024@163.com (F.Y.); wangln@ysfri.ac.cn (L.W.); lizt@ysfri.ac.cn (Z.L.); lill@ysfri.ac.cn (L.L.); xuyidan7@126.com (Y.X.); 3Key Laboratory for Sustainable Development of Marine Fisheries, Yellow Sea Fisheries Research Institute, Ministry of Agriculture and Rural Affairs, Qingdao 266071, China; 4Hainan Innovation Research Institute, Chinese Academy of Fishery Sciences, Sanya 572025, China

**Keywords:** *Epinephelus fasciatus* ♀ × *Plectropomus leopardus* ♂, embryonic development, mitochondrial genome, co-linearity, phylogenetic analyses

## Abstract

First construction of intergeneric hybrid grouper (*Epinephelus fasciatus* ♀ × *Plectropomus leopardus* ♂).

## 1. Introduction

Within the Serranidae family, both the black-edged grouper (*Epinephelus fasciatus*) and the leopard coral grouper (*Plectropomus leopardus*) inhabit tropical and subtropical coral reefs. Known for their bright orange-red colors, they are highly valued for their tasty meat, making them important species in aquaculture and ornamental fish markets. Morphologically, *E. fasciatus* features an orange-red body, with a reddish-brown upper head and back. Its sides display 5–6 broad horizontal bands, and the tips of its dorsal fin spines are black. This species is mainly found in the eastern coastal waters of the Indian Ocean extending to the central Pacific Ocean, including the waters around Taiwan, China, and the South China Sea [1]. In contrast, *P. leopardus* has a vivid red body adorned with small blue spots on its trunk and head [2]. It is primarily distributed in the southwestern Pacific Ocean, northern Australia, and the South China Sea [3]. *E. fasciatus* is a recently developed and valuable aquaculture species, with juvenile fish (5–8 cm) priced at 40 Yuan each at the factory gate, though supply does not meet demand [4]. However, *E. fasciatus* grows slowly and is a medium-to-small sized fish, with adults reaching only about 50 cm in length [5]. Under artificial farming conditions, one-year-old fish weigh just 58.89 ± 18.78 g, which limits large-scale cultivation and industrial promotion. On the other hand, *P. leopardus* is also a prized aquaculture species, capable of reaching a maximum adult length of 120 cm and weighing 525 g at one year old [6]. Nevertheless, as aquaculture expands, challenges such as shrinking breeding stock size, low survival rates of juveniles, poor stress tolerance, and frequent diseases have arisen, severely hindering sustainable grouper farming. To overcome these industry challenges, developing fast-growing, high-quality aquaculture germplasm through effective genetic improvement is crucial for advancing the grouper seed industry and the overall grouper aquaculture sector.

Distant hybridization breeding technology allows the combination of genomes from species that are genetically distant, producing hybrid offspring with highly diverse genotypes and phenotypes. Groupers, with more than 160 species, offer a valuable genetic resource for hybrid breeding studies. In China, grouper aquaculture produces 267,488 tons annually, generating an output value of 30 billion Yuan, with hybrid varieties making up over 70% of this production. Among the newly approved grouper varieties, three hybrids of *E. fuscoguttatus* ♀ × *E. lanceolatus* ♂, *E. moara* ♀ × *E. lanceolatus* ♂, and *E. fuscoguttatus* ♀ × *E. tukula* ♂ have been successfully developed. These hybrids exhibit notable hybrid vigor in growth, enhanced stress resistance, and higher survival rates. For example, the body weight of the 245-day-old *E. moara* ♀ × *E. lanceolatus* ♂ was 2.6 times greater than that of the maternal *E. moara* [7]. Research on *E. fuscoguttatus* ♀ × *E. tukula* ♂ revealed that 15-month-old hybrids grew 103% faster than the maternal *E. fuscoguttatus* and demonstrated better cold tolerance, as evidenced by significantly less feeding stoppage at semi-lethal temperatures compared to the parent strains [8].

Currently, interspecific hybridization dominates grouper crossbreeding efforts, while intergeneric hybridization studies are rare. For instance, the hybrid offspring of *Cromileptes altivelis* ♀ × *E. tukula* ♂ exhibits significant growth heterosis. At 330 days old, their body weight averages 220.50 ± 25.30 g, which is 1.55 times that of the maternal *C. altivelis*, and their total length is 23.57 ± 0.94 cm, 1.28 times longer than the maternal parent [9]. Similarly, the intergeneric hybrid offspring of *C. altivelis* ♀ × *E. lanceolatus* ♂ grow significantly faster than the maternal *C. altivelis*, with an absolute weight growth rate 1.6 times greater and a 4.70% higher meat yield compared to the paternal *E. lanceolatus* [10]. In addition, embryos from the intergeneric hybrid *E. fuscoguttatus* ♀ × *P. leopardus* ♂ develop normally. Comparative transcriptomic analysis of embryonic development between hybrid and parental embryos revealed that the Wnt signaling pathway plays a key role in regulating the formation of the embryonic dorsal-ventral axis [11].

Previous research has shown that selecting large-bodied, fast-growing males as paternal parents is essential for achieving optimal growth in grouper hybrids [12]. Consequently, the large and brightly colored *P. leopardus* was chosen as the paternal parent to enhance the growth rate of *E. fasciatus* through distant hybridization breeding, successfully producing viable hybrid offspring. By analyzing early embryonic development and comparing mitochondrial genomes, the feasibility of intergeneric hybridization in groupers was assessed to develop new hybrid germplasm with faster growth, better quality, and improved appearance. This study offers theoretical and technical support for distant hybridization breeding and the development of new grouper varieties.

## 2. Materials and Methods

### 2.1. Sampling and Observation

The experimental site and broodstock were supplied by Laizhou Ming Bo Aquaculture Co., Ltd. (Laizhou, China). Seven well-developed maternal *E. fasciatus* and four paternal *P. leopardus* with sperm motility exceeding 80% were selected for mixed fertilization. The sexually mature broodstock was administered injections comprising LHRH-A3 at dosage of 15 μg/kg and HCG at dosage of 300 IU/kg (Hubei Tusuo Technology Co., Ltd., Wuhan, China) to induce maturation and spawning. After 48 h of hormone treatment, female *E. fasciatus* with protruded genital pores were selected, and mature eggs were collected by gently pressing the abdomen into a dry plastic basin. Concurrently, high-motility sperm from the male *P. leopardus* were collected for artificial insemination. After a 5 min settling period, high-quality floating fertilized eggs were placed in micro-flow incubation barrels at 24.8 ± 0.5 °C, with salinity 28–30‰, and dissolved oxygen > 6.0 mg/L.

Before incubation, the fish eggs were soaked in 1 ppm povidone–iodine solution for 1 min. After 20 min, 100 floating fertilized eggs were collected from each incubation tank to count and record the fertilization rate, and this process was repeated three times. When the embryos developed to the tail bud stage, the eggs were transferred to nursery tanks for subsequent rearing, with an egg density of 8.0 g/m^3^, seawater temperature was maintained at 25 ± 1 °C, salinity at 28–30‰, dissolved oxygen at 6–8 mg/L, and water flow rate of 0.2 m^3^/h.

Thirty buoyant eggs were regularly taken from the hatching barrel, and an optical microscope (Olympus CX43, Yijingtong Optics Technology Co., Ltd., Shanghai, China) was used to photograph and measure the embryonic development of the new hybrid groupers. The time and developmental characteristics of each developmental stage were recorded. The timing for each stage was determined when two-thirds of the fertilized eggs had reached that particular phase.

### 2.2. DNA Extraction and Sequencing

Genomic DNA was extracted from the parental lines and hybrid using the DNA kit (OMEGA, Guangzhou, China), and the concentration and quality of the DNA were detected and analyzed with a Qubit 4.0 fluorometer and 1.0% agarose gel electrophoresis, respectively. Genomic libraries were sequenced using paired-end reads on the Illumina NovaSeq 6000 platform, and the sequencing data were processed with Trimmomatic v0.39. The primary workflow included removing adapter sequences from reads, trimming 5′-end bases containing non-AGCT nucleotides before clipping, trimming low-quality read ends (sequencing quality value < Q20), filtering out reads with N content ≥ 10%, and discarding short fragments < 75 bp after adapter removal and quality trimming.

### 2.3. Mitogenome Assembly and Gene Annotation

The mitochondrial genomes of the parent and hybrid were assembled using GetOrganelle version 1.7.5. This tool performs iterative extraction of target reads using the SEED database, then employs the SPAdes assembler to construct the genome. Sequences with sufficiently high coverage and long assembled lengths were chosen as candidates and aligned against the NCBI database to verify mitochondrial scaffolds. The confirmed sequences were then merged according to their overlapping regions. Using the parental reference genome, the starting point and orientation of the assembled mitochondrial sequence were established to produce the final mitochondrial genome sequence. The MITOS software version 1.1.3 was used to predict protein-coding genes (PCGs), tRNAs, and rRNAs within the mitochondrial genome. After redundancy removal and manual correction of the predicted genes, the start and stop codon positions of the mitochondrial genes were confirmed, yielding a highly accurate set of conserved genes. Finally, CGView online software visualized the genome compositions through a circular map.

The base composition and gene distribution of animal mitochondria were statistically analyzed and summarized, with the coding genes, rRNA, and tRNA arranged in genomic coordinate order. The mitochondrial genome distribution was visually displayed by identifying information such as gene length, gene interval, and codon composition. The cusp software (EMBOSS v6.6.0.0) was used to calculate the Relative Synonymous Codon Usage (RSCU) to assess codon preference.

### 2.4. Mitogenome Collinearity Analysis

Blastp alignment (e-value < 1 × 10^−5^) was performed against the protein sequences of the parental and hybrid mitochondrial genomes. For genes with multiple alignments in the database, only the best alignment result for each gene was retained. AliTV software version 1.0.6 was used to analyze the collinearity of the mitochondrial genomes.

### 2.5. Mitogenome Ka/Ks Analysis

To understand natural selection pressure during the evolution of *P. leopardus*, *E. fasciatus,* and their hybrids, gene sequences were compared using the MUSCLE v3.8.31, and then the KaKs Calculator 2.0 was used to calculate the non-synonymous (Ka) synonymous (Ks) ratio (Ka/Ks) with −c 11 (codon table), −m = MS (model selection based on AICc index).

### 2.6. Mitogenome Phylogeny

To evaluate the genomic evolutionary relationship of the hybrid germplasm, mitochondrial genome sequences of known pure and hybrid groupers were downloaded from the NCBI database. Nucleotide sequences of 13 coding genes from 19 grouper genomes were selected for multiple sequence alignment of shared single-copy genes using MUSCLE v3.8.31. Maximum Likelihood and Bayesian phylogenetic trees were constructed in PlyML v3.0 and MrBayes v3.2.6, respectively, and the final evolutionary tree was displayed using iTOL 3.4.3.

## 3. Results

### 3.1. Embryonic Development Observation and Mitochondrial Group Comparative Analysis

The fertilized eggs of the hybrid germplasm were transparent, spherical, and buoyant. At water temperatures of 24.8 ± 0.5 °C, the fertilized egg completed embryonic development in 28 h 55 min. Early embryonic development comprises six major stages: cleavage, blastula, gastrula, neurula, organogenesis, and hatching. The developmental characteristics of each stage are summarized in Table 1, and the main morphological features are illustrated in Figure 1.

The cleavage pattern of the hybrid embryo was incomplete discoid cleavage, with an egg diameter of 0.90 ± 0.02 mm. At 00:41 h after fertilization (HAF) (Figure 1a), a curved disk-shaped blastoderm began to form at the animal pole (Figure 1b). At 00:48 HAF, the embryo reached the cleavage stage, which included eight phases: the 2-cell, 4-cell, 8-cell, 16-cell, 32-cell, 64-cell, multicellular, and morula stages (Figure 1c–j). The blastomeres were divided into equal-sized cells from the 2-cell stage to the 32-cell stage. However, the dividing cells were irregular and unequal in size and began to overlap. At 03:05 HAF, the embryo entered the morula stage, with the appearance of a mulberry-shaped mass of cells. At 03:41 HAF, the embryos progressed to the blastula stage (Figure 1k,l), and at 05:53 HAF, the embryo’s development reached the gastrula stage (Figure 1m–o). At 10:50 HAF, the embryos entered the neurula stage, followed by the organogenesis stage at 13:13 HAF (Figure 1p–w) and the hatching stage at 27:14 HAF (Figure 1(z1)). Embryonic development was completed when more than two-thirds of the larvae had hatched at 28:55 h after fertilization (HAF) (Figure 1(z2)). The fertilization rate was 85.87 ± 5.22%, and the hatching rate was 70.37 ± 0.33%. The rate of deformities observed during embryonic development was 8.94 ± 1.57%. The newly hatched hybrid larvae had plump yolk sacs on their abdomens, with no pigment deposition on the body surface, and the notochord showed the physiological S-shaped curve. The total length of the newly hatched larvae was 2.05 ± 0.37 mm.

### 3.2. Mitochondrial Genome Composition

The mitochondrial genome sizes of the paternal *P. leopardus*, the maternal *E. fasciatus*, and their hybrid offspring were 16,753 bp, 16,570 bp, and 16,570 bp, respectively (Figure 2). A total of 37 known genes were separately identified in their mitochondrial genomes, including 13 PCGs, 22 tRNA genes, and two rRNA genes (Figure 2, Appendix A).

Among the 13 PCGs of the paternal genome, all genes started with ATG except for the start codon of *cox1*, which began with GTG. There were three stop codons for its PCGs, namely TAG, TAA, and T. For the maternal *E. fasciatus*, three start codons were found in the mitochondrial genes, where the *atp6* gene started with ATA, while *cox1* started with GTG, and all the other genes started with ATG. For its protein stop codons, only TAA and T were found. In the hybrid offspring, only three start codons (GTG, ATA, and ATG) and two stop codons (TAA and T) were found, and the mitochondrial composition was consistent with that of the maternal *E. fasciatus*. The shortest tRNA of 67 bp, *trnC* (GCA), and the longest tRNA of 78 bp, *trnS1* (GCT), were also identified in the paternal mitochondrial genome. The shortest tRNA of *E. fasciatus* was 67 bp (*trnC* (GCA)), while the longest tRNA was 75 bp (*trnL2* (TAA)). The corresponding results of the hybrid offspring were consistent with those of the maternal *E. fasciatus* (Appendix A).

The mitochondrial base compositions of the three groupers are shown in Table 2. The ratios of *P. leopardus* were 29.11% A, 27.92% T, 26.75% C, 16.22% G, with an AT bias of 57.03%, and a GC bias of 42.97%. The nucleotide composition of *E. fasciatus* revealed 28.73% A, 26.81% T, 28.32% C, 16.14% G, an AT bias of 55.55%, and a GC bias of 44.45%. The hybrid genome included 28.73% A, 26.82% T, 28.32% C, 16.13% G, an AT bias of 55.55%, and a GC bias of 44.45%. The base composition of the hybrid germplasm was largely consistent with that of *E. fasciatus*, with only slight differences in the T and G base content.

### 3.3. Analysis of Mitochondrial PCGs

Figure 3 shows the RSCU values of the PCGs in three grouper species. The hybrid offspring had 29 codons with an RSCU greater than 1, and the composition of these codons was entirely consistent with that of the maternal *E. fasciatus*, but significantly different from the paternal *P. leopardus*. The paternal *P. leopardus* had 31 preferred codons with RSCU values above 1, which is two more than both the hybrid offspring and the maternal *E. fasciatus*. Regarding the nucleotide distribution at the 3′ end of the codons, *P. leopardus* had 12 codons ending with A, 12 ending with C, and seven ending with T. In contrast, the hybrid offspring and maternal *E. fasciatus* showed a high level of similarity in this distribution: among the 29 codons with RSCU > 1 in *E. fasciatus*, 11 ended with A, 13 with C, and five with T, and the hybrid offspring fully inherited this pattern from the mother.

Furthermore, despite differences between parents and offspring in the number and types of preferred codons (RSCU > 1) among the three grouper species, they shared similar codon abundance patterns. Specifically, amino acids such as Ala, Arg, Gly, Leu1, Pro, Ser2, Thr, and Val had the highest codon abundance, while Trp was encoded by only one codon and showed the lowest abundance across all three species.

### 3.4. Collinearity Analysis

A collinearity comparison of the mitochondrial genomes of the hybrid and its parents was conducted based on their mitochondrial gene linkage relationships (Figure 4). The mitochondrial genomes of the three species were found to be largely consistent in both gene composition and arrangement. However, the hybrid was genetically closely related to the maternal *E. fasciatus*, which is consistent with the maternal inheritance pattern of vertebrates.

### 3.5. Ka/Ks Analysis

To assess the selective pressure on PCGs within the mitochondrial genome of the hybrid, the non-synonymous (Ka) and synonymous (Ks) substitution rates of each mitochondrial gene were determined (Figure 5). The results showed that the Ka/Ks ratio of all PCGs was less than 1, indicating that the mitochondrial genome underwent purifying selection. Excluding the nad6 gene, which exhibited a higher Ka/Ks ratio of 0.34, the Ka/Ks ratios of the other PCGs were all less than 0.10, suggesting that each gene experienced considerable negative selection pressure with high amino acid sequence conservation.

### 3.6. Phylogenetic Analysis

A phylogenetic analysis was conducted using the 13 PCGs sequences from 10 purebred grouper species (*Epinephelus*, *Cromileptes*, *Plectropomus*, and *Cephalopholis*), and their nine hybrid offspring (Figure 6). The analysis revealed that each hybrid offspring had a closer genetic relationship to its maternal parent, consistent with maternal inheritance patterns. Additionally, the genetic distance between the hybrid and the species from the maternal genus is shorter than that with the paternal *P. leopardus*. It is important to note that *Cromileptes* and *Epinephelus* share a relatively close genetic relationship, showing only slight differences between the genera.

## 4. Discussion

This study successfully bred intergeneric hybrid offspring between *P. leopardus* (♂) and *E. fasciatus* (♀) using hybridization breeding technology. At a water temperature of 24 ± 0.8 °C, the hybrid germplasm completed embryo development in 28 h 55 min, with a development rate that is faster than the maternal *E. fasciatus* (31 h 12 min) [13]. The total length of the newly hatched larvae was 2.05 ± 0.37 mm, significantly longer than that of the maternal *E. fasciatus* (1.44 ± 0.06 mm) [13], indicating that the hybrid germplasm exhibited growth heterosis in its early development, and providing theoretical support for the subsequent cultivation and promotion of this new hybrid germplasm. Similarly, the hybrid F1 of *E. moara* ♀ × *E. septemfasciatus* ♂ exhibited a greater body length than its parents *E. moara* ♀ and *E. septemfasciatus* ♂ [14]. Furthermore, the total length of the newly hatched larvae of *E. moara* ♀ × *E. lanceolatus* ♂ was 1.95 ± 0.06 mm, which was significantly longer than that of *E. moara* (1.64 ± 1.42 mm) or *E. lanceolatus* (1.53 ± 0.12 mm) [15]. The growth heterosis in response to hybridization breeding is crucial for the genetic improvement of the growth of *E. fasciatus*. This study represents the first successful breakthrough in cultivating viable intergeneric hybrid grouper (*E*. *fasciatus* ♀ × *P*. *leopardus* ♂). However, systematic growth comparison experiments have not yet been conducted, and relevant experiments will be supplemented in subsequent research.

The hybrid germplasm hatched earlier than the maternal *E. fasciatus* when incubated under the same conditions, indicating that the offspring developed embryonically at a faster rate, which suggests improved growth performance. The speed of embryonic development in groupers varies depending on incubation temperature. While the degree–hours model (calculated as water temperature multiplied by the number of hours elapsed) sums up temperature exposure, this two-parameter model uses the product of degree–hours to account for both acceleration and deceleration of development at high and low temperatures, providing more precise estimates of embryonic development [16]. The hybrid germplasm required fewer degree–hours than the maternal *E. fasciatus* but more than the paternal *P. leopardus* (Table 3). The incubation temperature for the paternal *P. leopardus* was relatively higher, about 30 °C, compared to 24.8 °C for both the hybrid germplasm and the maternal *E. fasciatus*. Temperature plays a crucial role in determining the speed of embryonic development [17]. Similarly to some fish species such as *E. malabaricus* and *E. septemfasciatus*, growth rates tend to increase with rising temperatures [18,19]. This observation may help in understanding species’ adaptability to water temperature and in optimizing temperature control for artificial breeding. Moreover, the hybrid germplasm, when incubated at the same relatively low temperature, showed initial potential to balance tolerance to low temperatures with efficient development. With further experimental data, this could offer a new approach for cultivating grouper seedlings under low-temperature conditions.

The mitochondrial genomes of *P. leopardus*, *E. fasciatus*, and their hybrid offspring were composed of 13 PCGs, 22 tRNA genes, and 2 rRNA genes. Most PCGs in the three mitochondrial genomes started with the typical ATG, which is common to many bony fish [20]. However, the GTG start codon of the *cox1* gene has been reported in a few groupers [21]. In addition, the *atp6* gene of most groupers usually starts with CTG or ATG. For example, in the hybrid *Hyporthodus septemfasciatus* ♀ × *E. moara* ♂, except for the *cox1* and *atp6*, which use GTG and CTG as the start codons, respectively, all other genes use ATG as the start codon [16]. In the hybrid *E. moara* ♀ × *E. tukula* ♂, except for *cox1* and *nd4*, which use GTG, *atp6* that uses CTG, and *nd3* that uses ATA, the remaining genes all use ATG [22]. Interestingly, the *atp6* gene of the current hybrid germplasm and the maternal *E. fasciatus* uses ATA as the start codon, which is uncommon in most bony fish.

Stop codon usage varies among genes. In the new hybrid germplasm, only two types of stop codons, TAA and the incomplete stop codon T, were detected in the mitochondrial genome, which is consistent with that of the maternal *E. fasciatus*. However, the paternal *P. leopardus* had three types of stop codons, including TAA, TAG, and T. The presence of incomplete stop codons is relatively common in fish mitochondrial genomes and can be completed as TAA during post-transcriptional polyadenylation of the mRNA [23]. This is a “correction mechanism” in gene expression that ensures that even if the stop codon on the DNA is incomplete, it can still be correctly identified, avoiding translational read-through.

The mitochondrial genomes of *P. leopardus*, *E. fasciatus*, and their hybrid offspring all showed a higher AT than GC bias. The G content of the three groupers was lower than 17%, indicating a strong anti-G bias [24]. Most of the base composition of the hybrid largely corresponds to that of the maternal species, further supporting the theory of mitochondrial maternal inheritance. However, differences in the composition of T and G between the hybrid offspring and the maternal *E. fasciatus* can serve as the basis for discrimination between the two.

The relative probability of a specific codon in the synonymous codons encoding the corresponding amino acid can reflect the degree of codon preference, where the RSCU value is typically used to evaluate the usage bias of synonymous codons [25,26]. The RSCU results of *P. leopardus*, *E. fasciatus* and their hybrid offspring all indicate that Ala (GCT, GCC, GCA, and GCG), Arg (CGT, CGC, CGA, and CGG), Gly (GGT, GGC, GGA, and GGG), Leu1 (CTT, CTC, CTA, and CTG), Pro (CCT, CCC, CCA, and CCG), Ser2 (TCT, TCC, and T), Thr (ACT, ACC, ACA, and ACG), and Val (GTT, GTC, GTA, and GTG) are the most abundant amino acids, while Trp (TGG) is rarely used. The PCGs tend to use A and C rather than T and G on the third codon, similar to other bony fish. In *E. bilobatus*, *E. maculatus*, and *E. longispinis*, the PCG codons ending with A and C are the most common, while those ending with T and G are the fewest [23]. In addition, the hybrid offspring were highly consistent with the maternal *E. fasciatus* in codon preference, suggesting that maternal inheritance has a robust impact on the offspring’s mitochondrial coding sequence. Moreover, the RSCU values of the three species were not all equal to 1, indicating that there are varying degrees of bias in their utilization of amino acids. The frequency of codon usage is linked to how strongly a gene is expressed; genes that are expressed very efficiently show a stronger bias in codon usage compared to those expressed less efficiently, and they typically favor specific synonymous codons [27].

Genomic collinearity refers to the genetic linkage relationship, which assesses the evolutionary scale and phylogenetic distance between different species. In molecular evolution, Ka/Ks is a commonly used metric for assessing selection pressure and evolutionary relationships between species [28]. In this study, the Ka/Ks ratio for all PCGs was less than 1, signifying that the mitochondrial genomes of both the hybrid and parents underwent purifying selection. Mitochondrial PCGs are essential for oxygen use and energy metabolism, which are critical for an organism’s survival and growth. The low Ka/Ks ratios observed in all PCGs indicate their functional preservation, reflecting an evolutionary approach to adapting to ecological niches. Among the 13 PCGs, the *nad6* gene exhibited a relatively higher Ka/Ks ratio (0.34), but it still fell under purifying selection (less than 1), implying it experienced weaker negative selection pressure. This observation is consistent with studies on vertebrate and mollusk mitochondrial genomes [29]. It also aligns with the more relaxed evolutionary constraints seen in other grouper species [22], suggesting that variations in this gene may play an important role in adaptive evolution by modulating energy metabolism [30]. Using mitochondrial *nad6* as an additional marker to study population variation in Lepidoptera revealed generally low DNA polymorphism [31]. This corresponds with the typically low genetic diversity found in Lepidoptera pest populations, emphasizing the role of *nad6* in population genetic diversity. Moreover, variations in the *nad6* gene have been found to significantly impact the development of organs like kernels in distantly related species such as maize, underscoring the gene’s importance in regulating growth and development [32].

Phylogenetic studies, along with other analyses, indicate that the new intergeneric hybrid germplasm is genetically closer to the maternal *E. fasciatus* and follows the typical pattern of mitochondrial inheritance from the mother. Moreover, the genetic distance between the hybrid and the maternal genus species is shorter compared to that with the paternal *P. leopardus*. Within the Serranidae family, mitochondrial genomes of hybrid groupers consistently exhibit maternal inheritance [33,34,35]. This strict maternal transmission maintains the stability of essential genes involved in energy metabolism, providing a genetic basis for the hybrids’ high survival rates and adaptability to their environment. Importantly, the similarity between the hybrid’s mitochondrial genome and that of its maternal parent correlates positively with the genetic distance between the parents: the greater the genetic distance, the more intact the maternal mitochondrial genes inherited by the hybrid. For instance, in interspecific hybridization, the COⅠ gene haplotype of *E. coioides* ♀ × *E. lanceolatus* ♂ shows 99.7–100% similarity with the maternal *E. coioides*, with only minor nucleotide variations [36]. In contrast, in intergeneric hybridization, the genetic distance between *Cromileptes* and *Epinephelus* is much larger. The mitochondrial genes of the hybrid offspring from *C. altivelis* ♀ × *E. tukula* ♂ exhibit 100% homology with the maternal *C. altivelis*, completely excluding paternal gene influence [35]. Furthermore, the closer genetic relatedness of *Cromileptes* and *Epinephelus* render them suitable for distant hybridization breeding. In aquaculture, the hybrid offspring from these two genera have a relatively high survival rate and exhibit prominent growth heterosis. For example, the body weight of *C. altivelis* ♀ × *E. tukula* ♂ at 330 days of age was 220.50 ± 25.30 g, which was 1.55 times that of the maternal *C. altivelis* [9]. Furthermore, the growth rate of *C. altivelis* ♀ × *E. lanceolatus* ♂ was significantly faster than that of the maternal *C. altivelis*, at the absolute value of 1.6 times higher [10].

## 5. Conclusions

This study showed for the first time that *E*. *fasciatus* ♀ × *P. leopardus* ♂ intergeneric hybrid populations can be successfully established. The fertilized eggs of the new germplasm completed their embryonic development within 28 h 55 min, with newly hatched larvae measuring 2.05 ± 0.37 mm in total length. The hybrid’s mitochondrial genome was 16,570 base pairs long, and analyses of collinearity, Ka/Ks ratios, and phylogenetics collectively indicated that the hybrid germplasm is more closely related to its maternal *E. fasciatus*, following a maternal inheritance pattern. These findings contribute to expanding the mitochondrial genome database of groupers and offer a critical reference for developing new grouper germplasm.

## Figures and Tables

**Figure 1 animals-15-03445-f001:**
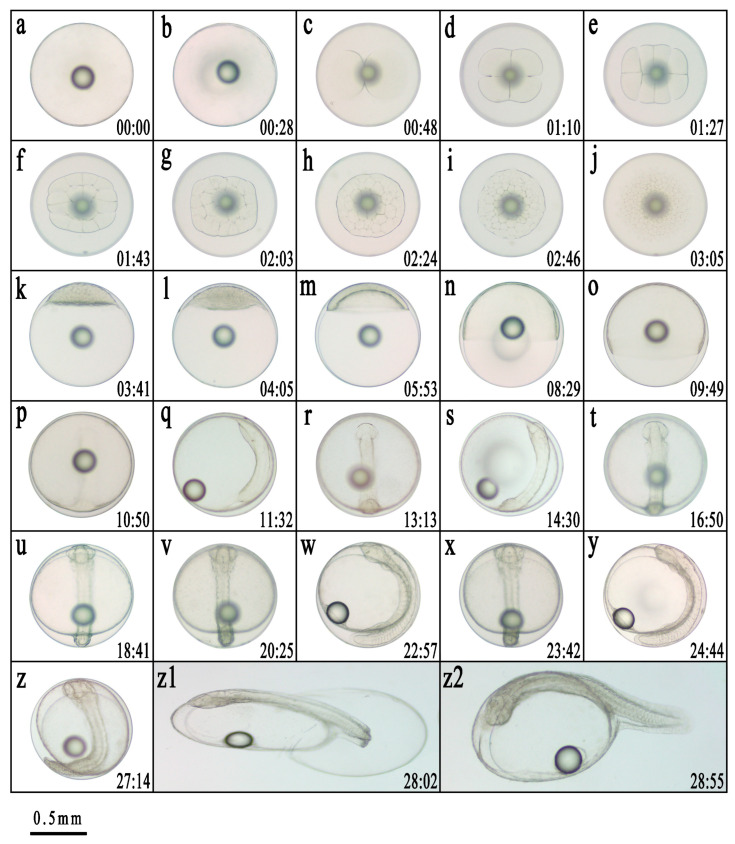
Embryonic development of *Epinephelus fasciatus* ♀ × *Plectropomus leopardus* ♂. (**a**): Fertilized egg; (**b**): Blastodisc formation; (**c**): 2-cell stage; (**d**): 4-cell stage; (**e**): 8-cell stage; (**f**): 16-cell stage; (**g**): 32-cell stage; (**h**): 64-cell stage; (**i**): Multi-cell stage; (**j**): morula stage; (**k**): High blastula stage; (**l**): Low blastula stage; (**m**): Early gastrula stage; (**n**): Middle gastrula stage; (**o**): Late gastrula stage; (**p**): Embryo body stage; (**q**): Closure of blastopore stage; (**r**): Optic capsule stage; (**s**): Muscle burl stage; (**t**): Otocyst stage; (**u**): Brain vesicle stage; (**v**): Heart stage; (**w**): Tail-bud stage; (**x**): Crystal stage; (**y**): Heart-beating stage; (**z**): Pre incubation stage; (**z1**): Hatching stage; (**z2**): Newly hatched larvae. The number in the lower right corner represents the time in hours and minutes after fertilization. The scale is 0.5 mm.

**Figure 2 animals-15-03445-f002:**
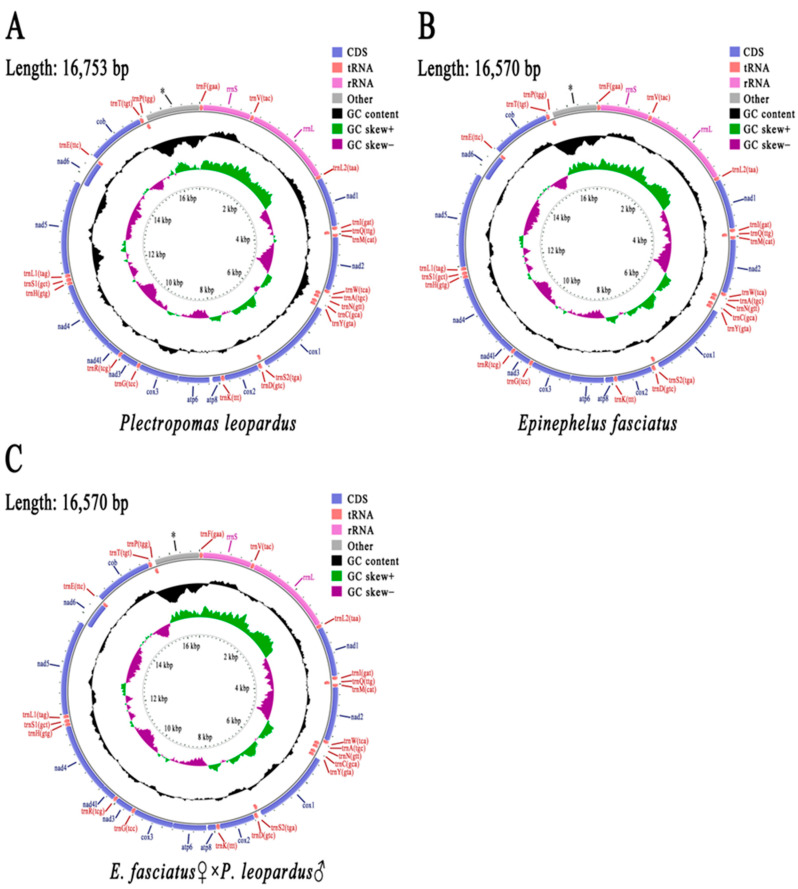
Mitochondrial genome maps of *Plectropomas leopardus* (**A**), *Epinephelus fasciatus* (**B**), and *E. fasciatus* ♀ × *P. leopardus* ♂ (**C**). The star * indicates A + T rich region containing origin of replication.

**Figure 3 animals-15-03445-f003:**
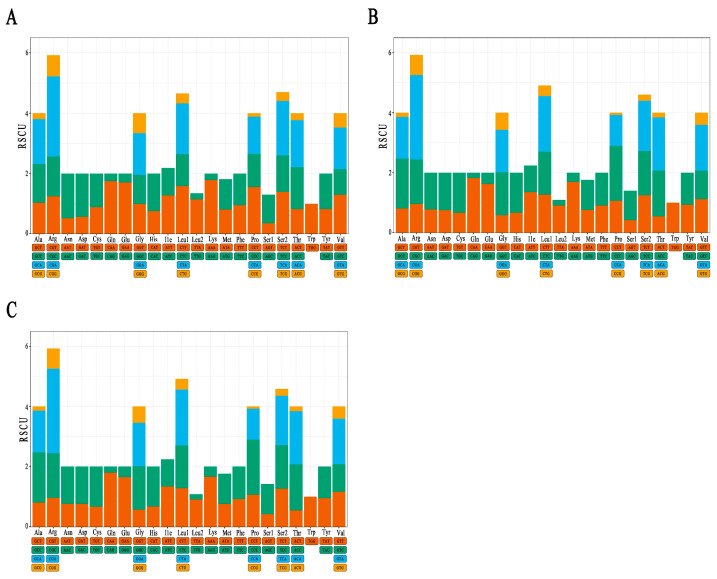
Relative synonymous codon usage (RSCU) of *Plectropomas leopardus* (**A**), *Epinephelus fasciatus* (**B**), and *E. fasciatus* ♀ × *P. leopardus* ♂ (**C**).

**Figure 4 animals-15-03445-f004:**
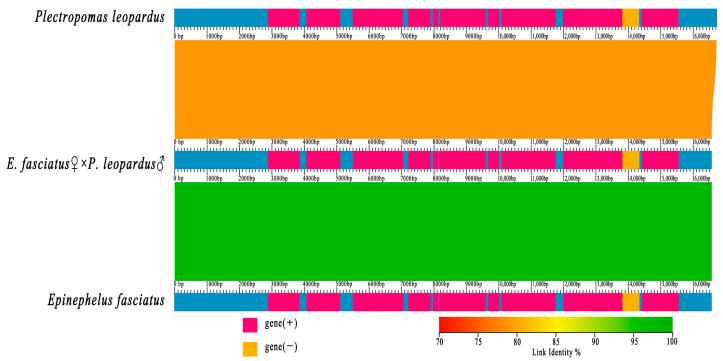
Mitochondrial whole genome alignment of *Plectropomas leopardus*, *Epinephelus fasciatus*, and *E. fasciatus* ♀ × *P. leopardus* ♂.

**Figure 5 animals-15-03445-f005:**
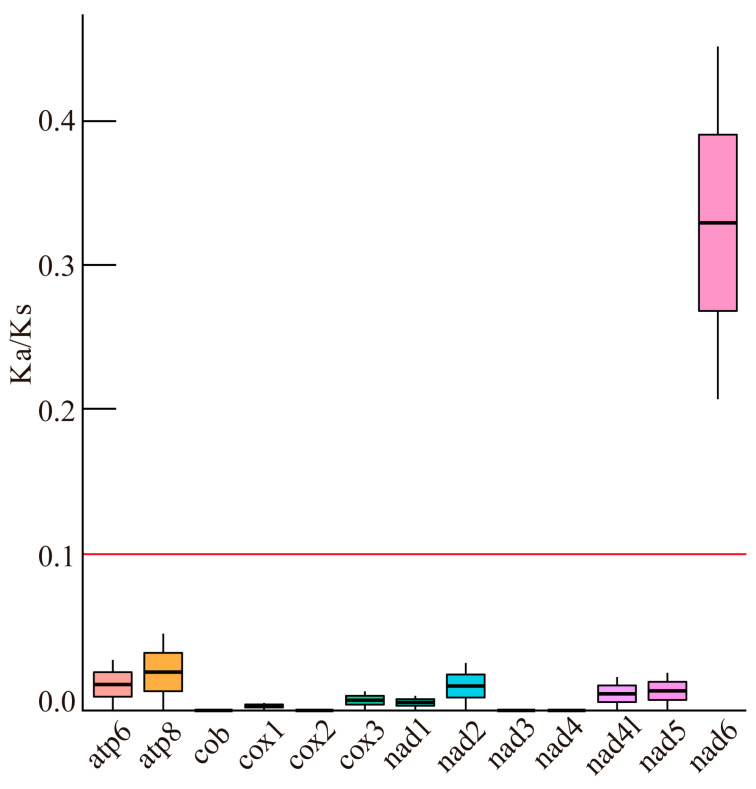
Ka/Ks values for 13 PCGs of *E. fasciatus* ♀ × *P. leopardus* ♂ compared to *Plectropomas leopardus* and *Epinephelus fasciatus.* Boxes in various colors indicate different mitochondrial genes.

**Figure 6 animals-15-03445-f006:**
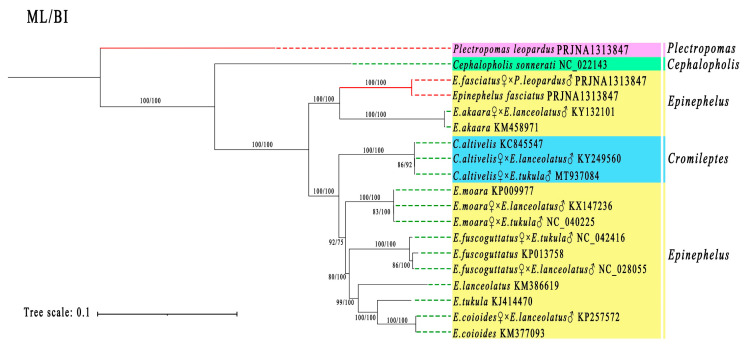
The combined maximum likelihood (ML) and Bayesian inference (BI) tree of 19 grouper species using 13 protein-coding genes (PCGs). Bootstrap values are shown at the base of each node. GenBank accession numbers follow scientific names. The species from the current study are highlighted with red dashed lines. Various background colors represent different genera of groupers.

**Table 1 animals-15-03445-t001:** Hybrid embryo development schedule of *Epinephelus fasciatus* ♀ × *Plectropomus leopardus* ♂.

Developmental Stage	Developmental Stage of Embryonic	Main Developmental Characteristics	Time After Fertilization
One-cell	fertilized egg	Spherical, with one oil ball	0
Cleavage	blastodisc formation stage	The blastoderm has formed, and when viewed from the side, it can be seen that the blastoderm protrudes like a cap	28 min
	2-cell stage	The first cleavage forms two cells	48 min
	4-cell stage	The second cleavage forms four cells	1 h 10 min
	8-cell stage	The third cleavage forms eight cells	1 h 27 min
	16-cell stage	The fourth cleavage forms 16 cells	1 h 43 min
	32-cell stage	The fifth cleavage forms 32 cells	2 h 03 min
	64-cell stage	The sixth cleavage formed 64 cells, and the division surface was rather disordered	2 h 24 min
	Multicellular stage	Continuous division leads to smaller cells and an increase in their number	2 h 46 min
	Morula stage	The cells are piled up in multiple layers, resembling mulberries in appearance	3 h 05 min
Blastula	high blastula stage	The blastocyst is tall and concentrated, and when viewed from the side, it appears as a high cap	3 h 41 min
	low blastula stage	The blastocyst becomes lower, and the cells are preparing to wrap towards the lower pole of the plant	4 h 05 min
Gastrula	early gastrula stage	One-third of the yolk is encapsulated under the germ layer	5 h 53 min
	middle gastrula stage	The embryo layer is subtracted from half of the yolk	8 h 29 min
	late gastrula stage	The embryo layer is subtracted to three-quarters of the yolk, the blastocyst becomes slender, and the embryo body is in the process of formation	9 h 49 min
Neurula	Embryonic formation stage	The embryonic body is formed with distinct contours	10 h 50 min
	Embryonic hole closure stage	The embryo layer is subcapsulated, and the embryo pore is completely closed	11 h 32 min
Organogenesis	Optic capsule stage	A pair of visual sacs appeared at the head of the embryo	13 h 13 min
	Muscle burl stage	Muscle segments appear in the middle of the embryo	14 h 30 min
	Otocyst stage	A pair of auditory sacs appeared at the posterior position of the visual sac in the head	16 h 50 min
	Brain vesicle stage	Brain vesicles appear between the two visual sacs	18 h 41 min
	Heart stage	The heart is formed on the ventral side with a clear outline	20 h 25 min
	Tail-bud stage	The tail of the embryo begins to separate from the yolk sac	22 h 57 min
	Crystal stage	Crystals appear in the eye of the embryo	23 h 42 min
	Heart-beating stage	The heart began to beat slightly and then gradually stabilized	24 h 44 min
Hatching	Early incubation stage	The embryo is twitching violently	27 h 14 min
	Incubation stage	The head emerges from the membrane first	28 h 02 min
	Newly hatched fry	The larvae hatch from the membrane	28 h 55 min
	Incubate for 48 h	The yolk sac shrinks and becomes smaller, the notochord becomes more obvious, the dorsal fin folds, the ventral fin folds, and the caudal fin membranes widen, pigmentation begins to form in the eyes, and the mouth can be slightly opened and closed	Total length is 2.35 ± 0.05 mm

**Table 2 animals-15-03445-t002:** Mitochondrial base composition of *Plectropomas leopardus*, *Epinephelus fasciatus*, and *E. fasciatus* ♀ × *P. leopardus* ♂.

Species	Region	Length (bp)	T%	C%	A%	G%	AT%	GC%
*Plectropomus leopardus*	Genome	16,753	27.92	26.75	29.11	16.22	57.03	42.97
	Protein_coding_genes	11,370	30.30	27.86	26.40	15.44	56.70	43.30
	First position	3790	28.71	28.23	26.70	16.36	55.41	44.59
	Second position	3790	31.58	28.39	24.85	15.17	56.44	43.56
	Third position	3790	30.61	26.97	27.65	14.78	58.26	41.74
	tRNA	1564	27.69	20.65	28.32	23.34	56.01	43.99
	rRNA	2656	22.06	23.43	33.43	20.97	55.50	44.50
*Epinephelus fasciatus*	Genome	16,570	26.81	28.32	28.73	16.14	55.55	44.45
	Protein_coding_genes	11,404	28.64	29.69	26.25	15.42	54.88	45.12
	First position	3801	24.49	26.99	26.57	21.94	51.06	48.93
	Second position	3801	33.88	29.83	21.91	14.36	55.8	44.2
	Third position	3801	27.52	32.25	30.25	9.97	57.77	42.22
	tRNA	1560	27.44	20.96	28.97	22.63	56.41	43.59
	rRNA	2653	21.56	25.52	32.42	20.51	53.98	46.02
*E. fasciatus* ♀ × *P. leopardus ♂*	Genome	*	26.82	*	*	16.13	*	*
	Protein_coding_genes	*	28.65	29.68	*	*	54.89	45.11
	First position	*	24.52	26.96	26.6	21.91	51.11	48.88
	Second position	*	33.83	29.88	*	*	55.74	44.25
	Third position	*	27.57	32.2	30.23	10	57.8	42.2
	tRNA	*	*	*	*	*	*	*
	rRNA	*	*	*	*	*	*	*

Note: The star * indicates that the values of the *E. fasciatus* ♀ × *P. leopardus* ♂ are consistent with those of the maternal *Epinephelus fasciatus*.

**Table 3 animals-15-03445-t003:** Comparison of degree–hours among different grouper species.

Species	Incubation Water Temperature (°C)	Hatching Time (hours)	Degree–Hours
*E. fasciatus* ♀ × *P. leopardus* ♂	24.8	28.55	708.04
*Epinephelus fasciatus*	24.8	31.12	771.78
*Plectropomas leopardus*	30	16.32	489.60

## Data Availability

The mitochondrial genome data presented in the study are deposited in the NCBI SRA repository (https://www.ncbi.nlm.nih.gov/ (accessed on 22 August 2025), accession number PRJNA1313847).

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
