# Peer review of "Comparative Analysis of Embryonic Development and Mitochondrial Genome of a New Intergeneric Hybrid Grouper (Epinephelus fasciatus ♀ × Plectropomus leopardus ♂)"

_animals, 2025, doi:10.3390/ani15233445_

Round 1
Reviewer 1 Report
Comments and Suggestions for Authors
The manuscript entitled " Comparative Analysis of Embryonic Development and Mitochondrial Genome of a New Intergeneric Hybrid Grouper (Epinephelus fasciatus ♀ ×Plectropomus leopardus ♂)" by Jiao et al. aim to produce and characterize a new intergeneric hybrid grouper derived from a cross between Epinephelus fasciatus (♀) and Plectropomus leopardus (♂), and to analyze its embryonic developmental process and mitochondrial genome features in comparison with its parental species.
The manuscript presents observations on the embryonic development and mitochondrial genome characterization of a novel intergeneric hybrid grouper. The topic is interesting and may have potential value for grouper hybrid breeding. However, the current version of the manuscript lacks sufficient novelty, methodological depth, and scientific interpretation to justify publication in Animals without major revision. The paper is descriptive, with minimal analytical insight or broader biological relevance. Extensive revision and clarification are required before reconsideration.
- The authors should clearly explain the main scientific question or hypothesis of the study, since confirming maternal mitochondrial inheritance is already well known in fish.
- Important information is missing, such as how many eggs were fertilized, the hatching rate, and the survival rate of the larvae. These details are necessary to judge whether the hybrid is healthy and viable. Please include data on fertilization, hatching, and survival rates, along with the number of replicates used in each test.
- The mitochondrial genome analysis is limited to basic sequence alignment and gene annotation, without any discussion of what these findings mean biologically. Please include functional or adaptive interpretations such as how specific mitochondrial gene features might relate to energy metabolism, growth performance, or hybrid adaptation.
- The authors should relate their genomic results (such as codon usage patterns and Ka/Ks ratios) to what they might mean in terms of evolution or function, for example, how these findings could explain hybrid growth advantages, adaptability, or mitochondrial efficiency. Please add a discussion linking these genomic indicators to possible roles in energy metabolism, stress tolerance, or enhanced growth performance of the hybrid.
- Some tables are redundant or overly long (e.g., Table 2 lists every gene position without highlighting key differences). Please summarize or move detailed lists to supplementary material.
- The manuscript needs major English editing to make the writing clearer, more concise, and easier to read. Many sentences are too long or awkwardly written
- The description of the hybridization and rearing process is incomplete. It does not include details such as where the broodstock came from, how many crosses were made, or what environmental conditions were controlled during the experiment. Please add clear information about the broodstock source, number of mating pairs or replicates, and the temperature, salinity, and other key conditions used during fertilization and incubation.
- The description of the hybridization and rearing process lacks information on broodstock origin, number of crosses, and environmental control conditions.
- The methods for mitochondrial genome assembly and annotation are standard but not fully transparent software versions and parameters should be specified more clearly.
- Some references (e.g., [13], [14]) appear incomplete or missing DOIs.
- Line 28–33: The Simple Summary should be rewritten in non-technical, concise English suitable for a general audience.
- Table 3: Combine repetitive values and highlight key comparative patterns (e.g., AT vs GC bias differences).
- Section 3.3: Include statistical or comparative remarks rather than only listing codon frequencies.
- Line 371–378 (Conclusions): Avoid overgeneralized claims such as “provide theoretical guidance for germplasm creation.” Be specific and modest about the actual contribution.
Author Response
R1’s comments:
The manuscript presents observations on the embryonic development and mitochondrial genome characterization of a novel intergeneric hybrid grouper. The topic is interesting and may have potential value for grouper hybrid breeding.
Response: Thank you very much for the positive comments.
Comments 1: However, the current version of the manuscript lacks sufficient novelty, methodological depth, and scientific interpretation to justify publication in Animals without major revision. The paper is descriptive, with minimal analytical insight or broader biological relevance. Extensive revision and clarification are required before reconsideration.
Response: Thank you for pointing this out. We concur with your observation and have included the corresponding details in the revised manuscript at the following lines: 15-16, 103-108, 114-120, 124-126, 139-147, 199-202, 267-282, 317-320, 341-367, 420-453, and 462-470.
Comments 2: The authors should clearly explain the main scientific question or hypothesis of the study, since confirming maternal mitochondrial inheritance is already well known in fish.
Response: Thanks. We agree with this comment. Therefore, we have added the relevant description as “Additionally, the genetic distance between the hybrid and the species from the maternal genus is shorter than that with the paternal P. leopardus.” in lines 317-318; “Importantly, the similarity between the hybrid’s mitochondrial genome and that of its maternal parent correlates positively with the genetic distance between the parents: the greater the genetic distance, the more intact the maternal mitochondrial genes inherited by the hybrid. For instance, in interspecific hybridization, the COâ… gene haplotype of E. coioides ♀ × E. lanceolatus ♂ shows 99.7%–100% similarity with the maternal E. coioides, with only minor nucleotide variations [36]. In contrast, in intergeneric hybridization, the genetic distance between Chromileptes and Epinephelus is much larger. The mitochondrial genes of the hybrid offspring from C. altivelis ♀ × E. tukula ♂ exhibit 100% homology with the maternal C. altivelis, completely excluding paternal gene influence [35].” in lines 444-453.
Comments 3: Important information is missing, such as how many eggs were fertilized, the hatching rate, and the survival rate of the larvae. These details are necessary to judge whether the hybrid is healthy and viable. Please include data on fertilization, hatching, and survival rates, along with the number of replicates used in each test.
Response: Thank you for bringing this to our attention. We concur with your observation and have included the corresponding details at the following lines: 21-24, 103-105, 114-120, and 199-202.
Comments 4: The mitochondrial genome analysis is limited to basic sequence alignment and gene annotation, without any discussion of what these findings mean biologically. Please include functional or adaptive interpretations such as how specific mitochondrial gene features might relate to energy metabolism, growth performance, or hybrid adaptation.
Response: Thanks. We agree with this comment. Therefore, we have added the relevant discussion in lines 420-435, and 436-453.
Comments 5: The authors should relate their genomic results (such as codon usage patterns and Ka/Ks ratios) to what they might mean in terms of evolution or function, for example, how these findings could explain hybrid growth advantages, adaptability, or mitochondrial efficiency. Please add a discussion linking these genomic indicators to possible roles in energy metabolism, stress tolerance, or enhanced growth performance of the hybrid.
Response: Thank you for pointing this out. Therefore, we have added the relevant discussion as “Mitochondrial PCGs are essential for oxygen use and energy metabolism, which are critical for an organism's survival and growth. The low Ka/Ks ratios observed in all PCGs indicate their functional preservation, reflecting an evolutionary approach to adapting to ecological niches. Among the 13 PCGs, the nad6 gene exhibited a relatively higher Ka/Ks ratio (0.34), but it still fell under purifying selection (less than 1), implying it experienced weaker negative selection pressure. This observation is consistent with studies on vertebrate and mollusk mitochondrial genomes [29]. It also aligns with the more relaxed evolutionary constraints seen in other grouper species [22], suggesting that variations in this gene may play an important role in adaptive evolution by modulating energy metabolism [30].” in lines 420-429; “This strict maternal transmission maintains the stability of essential genes involved in energy metabolism, providing a genetic basis for the hybrids' high survival rates and adaptability to their environment.” in lines 441-444.
Comments 6: Some tables are redundant or overly long (e.g., Table 2 lists every gene position without highlighting key differences). Please summarize or move detailed lists to supplementary material.
Response: Thank you for pointing this out. We have re-summarized the tables and moved Table 2 to the supplementary material.
Comments 7: The manuscript needs major English editing to make the writing clearer, more concise, and easier to read. Many sentences are too long or awkwardly written.
Response: Thank you for pointing this out. We have corrected all the sentences in the manuscript to ensure that the language is concise and clear. And mark the modified parts in red.
Comments 8: The description of the hybridization and rearing process is incomplete. It does not include details such as where the broodstock came from, how many crosses were made, or what environmental conditions were controlled during the experiment. Please add clear information about the broodstock source, number of mating pairs or replicates, and the temperature, salinity, and other key conditions used during fertilization and incubation.
Response: Thank you for pointing this out. We agree with this comment. Therefore, we have added the detailed description of the experiment in lines 102-120.
Comments 9: The description of the hybridization and rearing process lacks information on broodstock origin, number of crosses, and environmental control conditions.
Response: Thanks.We have added the detailed description of the experiment in lines 102-126.
Comments 10: The methods for mitochondrial genome assembly and annotation are standard but not fully transparent software versions and parameters should be specified more clearly.
Response: Thanks. We agree with this comment. Therefore, we have added the software versions and parameters in the revised manuscript, and relevant descriptions have been added in lines 139-147.
Comments 11: Some references (e.g., [13], [14]) appear incomplete or missing DOIs.
Response: Thank you for pointing this out. We have added the DOIs to all the references.
Comments 12: Line 28–33: The Simple Summary should be rewritten in non-technical, concise English suitable for a general audience.
Response: Thanks. We agree with this comment. Therefore, we have rewritten the content in the revised manuscript.
Comments 13: Table 3: Combine repetitive values and highlight key comparative patterns (e.g., AT vs GC bias differences).
Response: Thank you for pointing this out. We have corrected the relevant content in Table 3.
Comments 14: Section 3.3: Include statistical or comparative remarks rather than only listing codon frequencies.
Response: Thanks. We agree with this comment. Therefore, we have rewritten the contents in 3.3. Analysis of Mitochondrial PCGs
Comments 15: Line 371–378 (Conclusions): Avoid overgeneralized claims such as “provide theoretical guidance for germplasm creation.” Be specific and modest about the actual contribution.
Response: Thanks. We agree with this comment. Therefore, we have corrected them in lines 462–470.
Reviewer 2 Report
Comments and Suggestions for Authors
Q1. Scientific names should be italicized. Please confirm and revise the manuscript.
Q2. You stated that you observed 100 fertilized eggs during the developmental stages, but there's no detailed explanation of the timeline for each stage. For example, if 50 of the 100 blastocysts were observed, was the timeline calculated based on the number of blastocysts? Please provide a clear standard.
Q3. Please expand on the discussion to explain why the Ka/Ks ratio was lower than 1 in ND6, but was it consistently higher compared to other genes.
Author Response
R2’s comments:
Comments 1: Scientific names should be italicized. Please confirm and revise the manuscript.
Response: Thank you for pointing this out. We concur with your observation and have corrected the corresponding contents in the revised manuscript.
Comments 2: You stated that you observed 100 fertilized eggs during the developmental stages, but there's no detailed explanation of the timeline for each stage. For example, if 50 of the 100 blastocysts were observed, was the timeline calculated based on the number of blastocysts? Please provide a clear standard.
Response: Thank you for pointing this out. We have added the standard as “The timing for each stage was determined when two-thirds of the fertilized eggs had reached that particular phase.” in lines 124-126, and 199-200.
Comments 3: Please expand on the discussion to explain why the Ka/Ks ratio was lower than 1 in ND6, but was it consistently higher compared to other genes.
Response: Thank you for pointing this out. We agree with this comment. Therefore, we have added the relevant description in lines 423-435.
Reviewer 3 Report
Comments and Suggestions for Authors
The proposed manuscript entitled “Comparative Analysis of Embryonic Development and Mitochondrial Genome of a New Intergeneric Hybrid Grouper (Epinephelus fasciatus ♀ × Plectropomus leopardus ♂)” contributes to the study of developmental and molecular biology. I appreciate the practical part of the study, such as experimental crossing and animal husbandry.
The methodology is at a solid level and generates interesting data. However, the discussion part is currently presented in a very descriptive fashion rather than a critical analysis. While comparisons to published data are made, the manuscript lacks deeper interpretive elements such as novel hypotheses, evolutionary or phylogenetic scenarios, or an assessment of the findings' broader implications.
From my point of view, the study would be more impactful if the intention is to highlight the aquaculture potential of E. fasciatus × P. leopardus hybrids, particularly concerning heterosis. This should be explicitly stated and thoroughly explored. I suggest a clearer articulation of primary objectives.
Regarding embryonic development, the manuscript notes faster growth in hybrid embryos (28h55min) compared to the maternal species E. fasciatus (31h12min) (lines 291-300). This observation is a significant point of interest. However, a direct comparison with the paternal species’ embryonic development is absent, which limits the ability to assess the extent of any heterotic effect. Moreover, the intensity of growth can be influenced by many external factors and thus the best solution would be direct comparison of embryonic development in studied hybrids with their parents. These comparisons have not been performed in the proposed study. In addition, the discussion would benefit from addressing whether this accelerated growth persists into post-embryonic developmental stages, if such data are available.
The molecular analysis of the mitochondrial genome, while technically sound, concludes that the results are consistent with known maternal inheritance patterns in vertebrates. This finding does not appear to add substantial novel value or generate new insights beyond confirming that the results are consistent with the maternal inheritance patterns in vertebrates. The similar results should have been obtained if the authors easily amplify selected mitochondrial and nuclear genes using PCR and subsequently compare sequences among individuals.
Taken into consideration together:
1. The central hypothesis and goal of the study need clearer definition.
2. The manuscript lacks direct comparison between parents and hybrids to fully estimate heterotic effect.
Therefore, I am not convinced that the manuscript in the current form meets the impact and novelty requirements of the Animal journal and will be of interest to the journal’s intended readership.
Author Response
R3’s comments:
The proposed manuscript entitled “Comparative Analysis of Embryonic Development and Mitochondrial Genome of a New Intergeneric Hybrid Grouper (Epinephelus fasciatus ♀ × Plectropomus leopardus ♂)” contributes to the study of developmental and molecular biology. I appreciate the practical part of the study, such as experimental crossing and animal husbandry. The methodology is at a solid level and generates interesting data.
Response: Thank you very much for the positive comments.
Comments 1: However, the discussion part is currently presented in a very descriptive fashion rather than a critical analysis. While comparisons to published data are made, the manuscript lacks deeper interpretive elements such as novel hypotheses, evolutionary or phylogenetic scenarios, or an assessment of the findings' broader implications.
Response: Thank you for pointing this out. We concur with your observation and have corrected the corresponding discussion as “Additionally, the genetic distance between the hybrid and the species from the maternal genus is shorter than that with the paternal P. leopardus.” in lines 317-318; “Importantly, the similarity between the hybrid’s mitochondrial genome and that of its maternal parent correlates positively with the genetic distance between the parents: the greater the genetic distance, the more intact the maternal mitochondrial genes inherited by the hybrid. For instance, in interspecific hybridization, the COâ… gene haplotype of E. coioides ♀ × E. lanceolatus ♂ shows 99.7%–100% similarity with the maternal E. coioides, with only minor nucleotide variations [36]. In contrast, in intergeneric hybridization, the genetic distance between Chromileptes and Epinephelus is much larger. The mitochondrial genes of the hybrid offspring from C. altivelis ♀ × E. tukula ♂ exhibit 100% homology with the maternal C. altivelis, completely excluding paternal gene influence [35].” in lines 444-453.
Comments 2: From my point of view, the study would be more impactful if the intention is to highlight the aquaculture potential of E. fasciatus × P. leopardus hybrids, particularly concerning heterosis. This should be explicitly stated and thoroughly explored. I suggest a clearer articulation of primary objectives.
Response: Thanks. We agree with this comment. Therefore, we have corrected them in lines: 15-16, 18-24, 199-202, 341-367, and 462-465.
Comments 3: Regarding embryonic development, the manuscript notes faster growth in hybrid embryos (28h55min) compared to the maternal species E. fasciatus (31h12min) (lines 291-300). This observation is a significant point of interest. However, a direct comparison with the paternal species’ embryonic development is absent, which limits the ability to assess the extent of any heterotic effect. Moreover, the intensity of growth can be influenced by many external factors and thus the best solution would be direct comparison of embryonic development in studied hybrids with their parents. These comparisons have not been performed in the proposed study. In addition, the discussion would benefit from addressing whether this accelerated growth persists into post-embryonic developmental stages, if such data are available.
Response: Thank you for pointing this out. We agree with this comment. Therefore, we have added the corresponding description as “E. fasciatus is a recently developed and valuable aquaculture species, with juvenile fish (5-8 cm) priced at 40 Yuan each at the factory gate, though supply does not meet demand [4]. However, E. fasciatus grows slowly and is a medium-to-small sized fish, with adults reaching only about 50 cm in length [5]. Under artificial farming conditions, one-year-old fish weigh just 58.89 ± 18.78 g, which limits large-scale cultivation and industrial promotion.” in lines 49-55; “Previous research has shown that selecting large-bodied, fast-growing males as paternal parents is essential for achieving optimal growth in grouper hybrids [12]. Consequently, the large and brightly colored P. leopardus was chosen as the paternal parent to enhance the growth rate of E. fasciatus through distant hybridization breeding, successfully producing viable hybrid offspring.” in lines 90-94; “This study successfully bred intergeneric hybrid offspring between P. leopardus (♂) and E. fasciatus (♀) using hybridization breeding technology. At a water temperature of 24 ± 0.8℃, the hybrid germplasm completed embryo development in 28h55min, with a development rate that is faster than the maternal E. fasciatus (31h12min) [13]. The total length of the newly hatched larvae was 2.05 ± 0.37 mm, significantly longer than that of the maternal E. fasciatus (1.44 ± 0.06 mm) [13], indicating that the hybrid germplasm exhibited growth heterosis in its early development, and providing theoretical support for the subsequent cultivation and promotion of this new hybrid germplasm.” in lines 327-334; “This study represents the first successful breakthrough in cultivating viable intergeneric hybrid grouper (E. fasciatus ♀ × P. leopardus ♂). However, systematic growth comparison experiments have not yet been conducted, and relevant experiments will be supplemented in subsequent research.” in lines 340-343. The manuscript seeks to enhance the growth rate of the maternal E. fasciatus by leveraging the fast growth characteristics of the paternal P. leopardus through hybrid breeding. This effect has already been noted during the embryonic development phase. Further breeding assessments will be detailed in future research.
Comments 4: The molecular analysis of the mitochondrial genome, while technically sound, concludes that the results are consistent with known maternal inheritance patterns in vertebrates. This finding does not appear to add substantial novel value or generate new insights beyond confirming that the results are consistent with the maternal inheritance patterns in vertebrates. The similar results should have been obtained if the authors easily amplify selected mitochondrial and nuclear genes using PCR and subsequently compare sequences among individuals.
Response: Thank you for pointing this out. We concur with your observation and have added the corresponding description as “Additionally, the genetic distance between the hybrid and the species from the maternal genus is shorter than that with the paternal P. leopardus.” in lines 317-318; “Importantly, the similarity between the hybrid’s mitochondrial genome and that of its maternal parent correlates positively with the genetic distance between the parents: the greater the genetic distance, the more intact the maternal mitochondrial genes inherited by the hybrid. For instance, in interspecific hybridization, the COâ… gene haplotype of E. coioides ♀ × E. lanceolatus ♂ shows 99.7%–100% similarity with the maternal E. coioides, with only minor nucleotide variations [36]. In contrast, in intergeneric hybridization, the genetic distance between Chromileptes and Epinephelus is much larger. The mitochondrial genes of the hybrid offspring from C. altivelis ♀ × E. tukula ♂ exhibit 100% homology with the maternal C. altivelis, completely excluding paternal gene influence [35].” in lines 444-453; “The low Ka/Ks ratios observed in all PCGs indicate their functional preservation, reflecting an evolutionary approach to adapting to ecological niches. Among the 13 PCGs, the nad6 gene exhibited a relatively higher Ka/Ks ratio (0.34), but it still fell under purifying selection (less than 1), implying it experienced weaker negative selection pressure. This observation is consistent with studies on vertebrate and mollusk mitochondrial genomes [29]. It also aligns with the more relaxed evolutionary constraints seen in other grouper species [22], suggesting that variations in this gene may play an important role in adaptive evolution by modulating energy metabolism [30].” in lines 421-429.
Comments 5: The central hypothesis and goal of the study need clearer definition.
Response: Thank you for pointing this out. We concur with your observation and have added the corresponding description as “To develop superior grouper aquaculture varieties, in this study, a intergeneric hybrid breed was constructed by crossing a male Plectropomus leopardus, with a female parent Epinephelus fasciatus. Here, we researched embryonic development and mitochondrial composition of the new hybrid germplasm.” in lines 18-21.
Comments 6: The manuscript lacks direct comparison between parents and hybrids to fully estimate heterotic effect. Therefore, I am not convinced that the manuscript in the current form meets the impact and novelty requirements of the Animal journal and will be of interest to the journal’s intended readership.
Response: Thank you for pointing this out. We agree with this comment. Therefore, we have added the corresponding description as “The hybrid germplasm hatched earlier than the maternal E. fasciatus when incubated under the same conditions, indicating that the offspring developed embryonically at a faster rate, which suggests improved growth performance. The speed of embryonic development in groupers varies depending on incubation temperature. While the degree–hours model (calculated as water temperature multiplied by the number of hours elapsed) sums up temperature exposure, this two-parameter model uses the product of degree–hours to account for both acceleration and deceleration of development at high and low temperatures, providing more precise estimates of embryonic development [21]. The hybrid germplasm required fewer degree–hours than the maternal E. fasciatus but more than the paternal P. leopardus (Table 3). The incubation temperature for the paternal P. leopardus was relatively higher, about 30°C, compared to 24.8°C for both the hybrid germplasm and the maternal E. fasciatus. Temperature plays a crucial role in determining the speed of embryonic development [16]. Similar to some fish species such as E. malabaricus and E. septemfasciatus, growth rates tend to increase with rising temperatures [17, 18]. This observation may help in understanding species’ adaptability to water temperature and in optimizing temperature control for artificial breeding. Moreover, the hybrid germplasm, when incubated at the same relatively low temperature, showed initial potential to balance tolerance to low temperatures with efficient development. With further experimental data, this could offer a new approach for cultivating grouper seedlings under low-temperature conditions.” in lines 345-364. The manuscript aims to improve the growth rate of the maternal E. fasciatus by utilizing the rapid growth traits of the paternal P. leopardus through hybrid breeding. This effect has already been observed during the embryonic development stage. The study initially created a new intergeneric hybrid grouper and carried out a systematic investigation of their embryonic development and mitochondrial composition. This research presents a degree of novelty and aligns with the standards of the Animal journal.
Round 2
Reviewer 1 Report
Comments and Suggestions for Authors
Based on scientific consideration, authors had improved the manuscript.